# Dinucleotide biases in RNA viruses that infect vertebrates or invertebrates

Diego Forni,[1] Uberto Pozzoli,[1] Rachele Cagliani,[1] Mario Clerici,[2,3] Manuela Sironi[1]

**ABSTRACT** CpG and UpA dinucleotides are under-represented in vertebrate genomes, whereas most invertebrates only show a bias against UpA. RNA viruses are thought to have evolved genomes that resemble the dinucleotide composition of their hosts, possibly to avoid restriction by the zinc-finger antiviral protein (ZAP). By performing a comprehensive analysis of RNA viruses, we show that, whereas UpA dinucleotides are similarly under-represented irrespective of viral genome composition or host, important differences are observed for CpG. The tendency for vertebrate-infecting viruses to have stronger CpG bias than invertebrate-infecting viruses is not universal. Rather, it is mainly driven by single-stranded (ss) RNA(+) viruses. Conversely, ssRNA(−) viruses have a dinucleotide composition that is unrelated to the host clade. Also, these viruses, especially those in the order *Bunyavirales*, are extremely CpG-depleted. By focusing on specific viral families, we also show that, even for vertebrate ssRNA(+) viruses, ZAP is unlikely to be a driver of CpG depletion. Consistently, CpG dinucleotides tend to be preferentially depleted in A/U-rich contexts in both vertebrate- and invertebrate-infecting viruses. Finally, within the same viral genomes, individual viral open reading frames (ORFs) can display different CpG content. Analysis of SARS-CoV-2 revealed a remarkable depletion of CpG dinucleotides in ORF1ab and S, but not in N and M. Thus, these results do not support the view that an adaptive shift for CpG depletion in the SARS-CoV-2 lineage occurred as an innate immunity evasion strategy. Our data provide a better understanding of viral evolution and inform approaches based on the modulation of CpG to generate attenuated viruses.

**IMPORTANCE** Akin to a molecular signature, dinucleotide composition can be exploited by the zinc-finger antiviral protein (ZAP) to restrict CpG-rich (and UpA-rich) RNA viruses. ZAP evolved in tetrapods, and it is not encoded by invertebrates and fish. Because a systematic analysis is missing, we analyzed the genomes of RNA viruses that infect vertebrates or invertebrates. We show that vertebrate single-stranded (ss) RNA(+) viruses and, to a lesser extent, double-stranded RNA viruses tend to have stronger CpG bias than invertebrate viruses. Conversely, ssRNA(−) viruses have similar dinucleotide composition whether they infect vertebrates or invertebrates. Analysis of ssRNA(+) viruses that infect mammals, reptiles, and fish indicated that ZAP is unlikely to be a major driver of CpG depletion. We also show that, compared to other coronaviruses, the genome of SARS-CoV-2 is not homogeneously CpG-depleted. Our study provides new insights into virus evolution and strategies for recoding RNA virus genomes.

**KEYWORDS** RNA virus, CpG dinucleotide, UpA dinucleotide, zinc-finger antiviral protein

The genomes of cellular organisms display important dinucleotide composition biases. In vertebrates, the CpG dinucleotide shows the strongest bias, being highly under-represented both in nuclear and mitochondrial genomes (1–4). Conversely, the

Address correspondence to Manuela Sironi, manuela.sironi@lanostrafamiglia.it.

Diego Forni and Uberto Pozzoli contributed equally to this article. The order of names for first authors is alphabetical.

The authors declare no conflict of interest.

See the funding table on p. 16.

tendency to avoid CpG dinucleotides is not observed in the majority of invertebrates (3, 5–9). On the other hand, TpA dinucleotides are under-represented in virtually all living organisms, particularly in sequences expressed as RNA in the cytoplasm (3, 4, 7, 10). Different hypotheses have been proposed to explain such biases. The depletion of CpG dinucleotides in vertebrates is thought to be at least partially due to cytosine methylation (3, 5, 11), although this does not explain the bias in mitochondrial genomes (3). In the case of TpA (UpA in RNA), one of the most supported explanations is the preferential cleavage of UpA dinucleotides by cytosolic RNases (10).

Irrespective of the underlying mechanisms accounting for these biases in animal genomes, nucleotide composition is akin to a molecular signature, which can be exploited for the recognition of non-self nucleic acids. This is central to sense and counteract infecting agents, most notably viruses. Thus, whereas toll-like receptor 9 (TLR9) recognizes double-strand (ds) DNA with non-methylated CpG dinucleotides, the zinc-finger antiviral protein (ZAP) specifically binds CpG dinucleotides in single-stranded (ss) RNA (12–14). ZAP evolved in tetrapods (9), is expressed in the cytoplasm, and was shown to restrict RNA viruses and retroviruses carrying a high proportion of CpG (15–21). Recently, ZAP was also shown to restrict viruses with elevated frequencies of UpA dinucleotides (15, 19).

It is generally considered that to avoid restriction by ZAP (and possibly by other cellular proteins), RNA viruses have evolved genomes that resemble the genomic dinucleotide composition of their hosts (7, 22–25). For instance, Simmonds and co-workers analyzed the representation of CpG dinucleotides in the genomes of RNA and small DNA viruses that infect mammals and insects (which do not possess ZAP) (7). They found no CpG depletion among insect viruses. Conversely, mammalian RNA viruses with single-stranded genomes and reverse-transcribing viruses, but not dsRNA viruses, showed CpG suppression. Specifically, CpG depletion in these viruses was related to the G + C composition of their genomes. The authors thus concluded that mammal-infecting RNA viruses that expose their genetic material to the cytoplasm are subject to selection against CpG. This hypothesis is further supported by observations in flaviviruses (order *Amarillovirales*), whereby viruses that replicate only in insects have higher CpG content than viruses that alternate between vertebrate and invertebrate hosts (23). Also, the artificial increase of CpG content in the Zika virus (family *Flaviviridae*) attenuates infection in mammalian cells but increases viral titers in insect cells (26). Indeed, the generation of viruses with elevated CpG content was shown to represent a promising strategy to obtain attenuated strains for vaccination purposes (20, 27–31).

Despite these observations, a survey of animal RNA viruses indicated that the dinucleotide composition of their genomes more closely reflects viral family than host associations (32).

Recently, large-scale metagenomic approaches have revealed an enormous diversity of RNA viruses hosted by invertebrates. Most of these viruses show phylogenetic relationships to known genera or families of vertebrate viruses (33). Here, we performed a comprehensive analysis of RNA viruses that infect vertebrates or invertebrates, with the aim of identifying the determinants of biases against CpG and UpA dinucleotides.

## RESULTS

### Dinucleotide biases and correlation with G + C content depend on host group

We assembled a data set of 4,144 animal virus genomes or genome segments (Table S1). In particular, we obtained 2,308 sequences of invertebrate viruses from a large-scale metatranscriptomic project that analyzed more than 220 species (mostly arthropods) (33). The remaining 1,836 genomes or genome segments were retrieved from the International Committee on Taxonomy of Viruses (ICTV) virus metadata resource by including viruses that infect vertebrate hosts but excluding dual-host viruses. Reverse-transcribing viruses were also excluded because they have no representatives in the invertebrate virus data set. To investigate dinucleotide biases, we calculated the observed/expected ratio for CpG and UpA dinucleotides. As a comparison, the ratios of

GpC and ApU were also calculated. Specifically, the expected dinucleotide frequency in a sequence is simply the product of the frequencies of the contributing nucleotides. Thus, ratios higher or lower than 1 indicate that a dinucleotide is over- or under-represented, respectively. In particular, ratios lower than 0.78 and higher than 1.23 are generally considered to define significant depletion and enrichment (11, 34).

Analysis of vertebrate and invertebrate viruses indicated that CpG dinucleotides are strongly under-represented, especially in vertebrate viruses, which have a median ratio lower than 0.78. The ratio of CpG (rCpG) was characterized by a wide variability among viruses (Fig. 1A). UpAs were also under-represented, but at similar levels in vertebrate and invertebrate viruses. No under- or over-representation was detected for GpC and ApU dinucleotides, as expected (Fig. 1A).

Previous analyses of vertebrate genomes and viruses detected a positive correlation between rCpG and G + C content. A negative correlation with G + C content was instead detected for rUpA (4, 7, 35, 36). Both effects were shown to be at least partially explained by the selective depletion of CpG dinucleotides (36). We thus tested whether rCpG and rUpA correlated with G + C content in our data set of viruses infecting vertebrates and invertebrates. To do so, we fitted linear models with rCpG (or rUpA) as the dependent variable and G + C content and host group as independent variables. In additional models, the interaction between the independent variables was also fitted in the regression. The best model (with or without interaction) was selected using likelihood ratio tests. For rCpG, the model with interaction resulted to be preferred over that without interaction. Strong and significant effects were observed for both G + C content and the host group (Table S2; Fig. 1B). Also, the trajectories of the regression lines differed depending on the host group, with G + C content explaining a larger proportion of rCpG variation in vertebrate viruses (Fig. 1B). Qualitatively similar results were obtained for UpA. However, the effect of the host was definitely weaker than in the case of rCpG, and the slopes of the regression lines were much more similar (Fig. 1B).

Overall, these results indicate that CpG dinucleotides are more depleted and more strongly correlated to G + C content in vertebrate than in invertebrate viruses. UpAs show similar under-representation and dependency on G + C content in viruses infecting hosts from the two groups.

## Dinucleotide biases vary depending on host, genome compositions, and virus order

Previous analyses on smaller data sets showed that viruses with dsRNA genomes tend to display a weaker bias against CpG dinucleotides (7). Positive-sense ssRNA viruses [ssRNA(+)], negative-strand ssRNA viruses [ssRNA(−)], and dsRNA viruses are differentially represented in the vertebrate and invertebrate virus data sets (Fig. 1C). For instance, ssRNA(+) viruses account for ~50% and ~61% of vertebrate and invertebrate virus genomes/segments, respectively. We thus analyzed rGpC and rUpA against G + C content and host group for the three genome categories. Linear regressions with and without interactions were applied, as described above. The results of the model selection, as well as model coefficients, are summarized in Table S2; Fig. 2.

For rCpG, a minimal effect of the host group was evident for dsRNA and ssRNA(−) viruses. Conversely, ssRNA(+) viruses showed extremely different rCpG levels and markedly different slopes depending on the host group (Fig. 2). Thus, ssRNA(+) viruses, which are the most abundant in the data set, seem to explain a large portion of the differences between vertebrate and invertebrate viruses in the whole data set (Fig. 1B). The analysis also confirmed that dsRNA viruses have, on average, high rCpG, whereas ssRNA(−) viruses tend to display the lowest ratios (Fig. 2).

Concerning rUpA, no major differences between vertebrate and invertebrate viruses were evident. Also, dsRNA, ssRNA(−), and ssRNA(+) viruses displayed comparable biases against rUpA and similar slopes of regression lines (Fig. 2).

The major differences in the rGpC of viruses with different genome composition and infecting distinct hosts prompted us to determine how viruses in specific orders

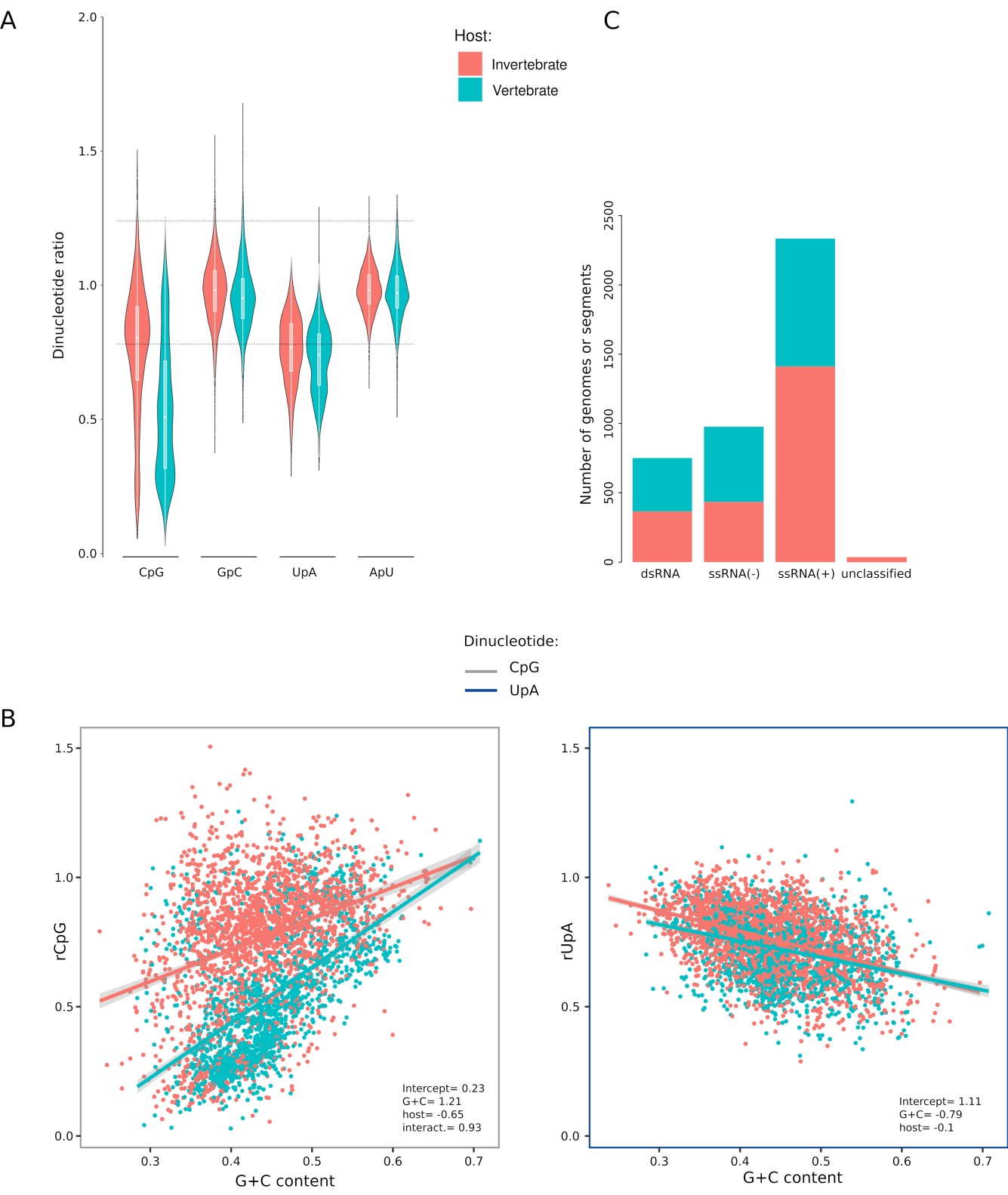

**FIG 1** Dinucleotide representation in viruses that infect vertebrates and invertebrates. (A) Violin plots with boxplot the observed/expected ratio (*r*) for CpG, GpC, UpA, and ApU. The horizontal hatched lines correspond to ratios of 0.78 and 1.23, which generally define significant depletion and enrichment (11, 34). (B) Comparison of rCpG (left) and rUpA (right) for RNA viruses that infect vertebrates and invertebrates. Regression lines of rCpG or rUpA on G + C content are shown with confidence intervals. The results of the linear models are included in each plot. Specifically, only significant model coefficients are reported (full model results and likelihood ratio tests are available in Table S2). (C) Distribution of viral genomes/genome segments in the data set.

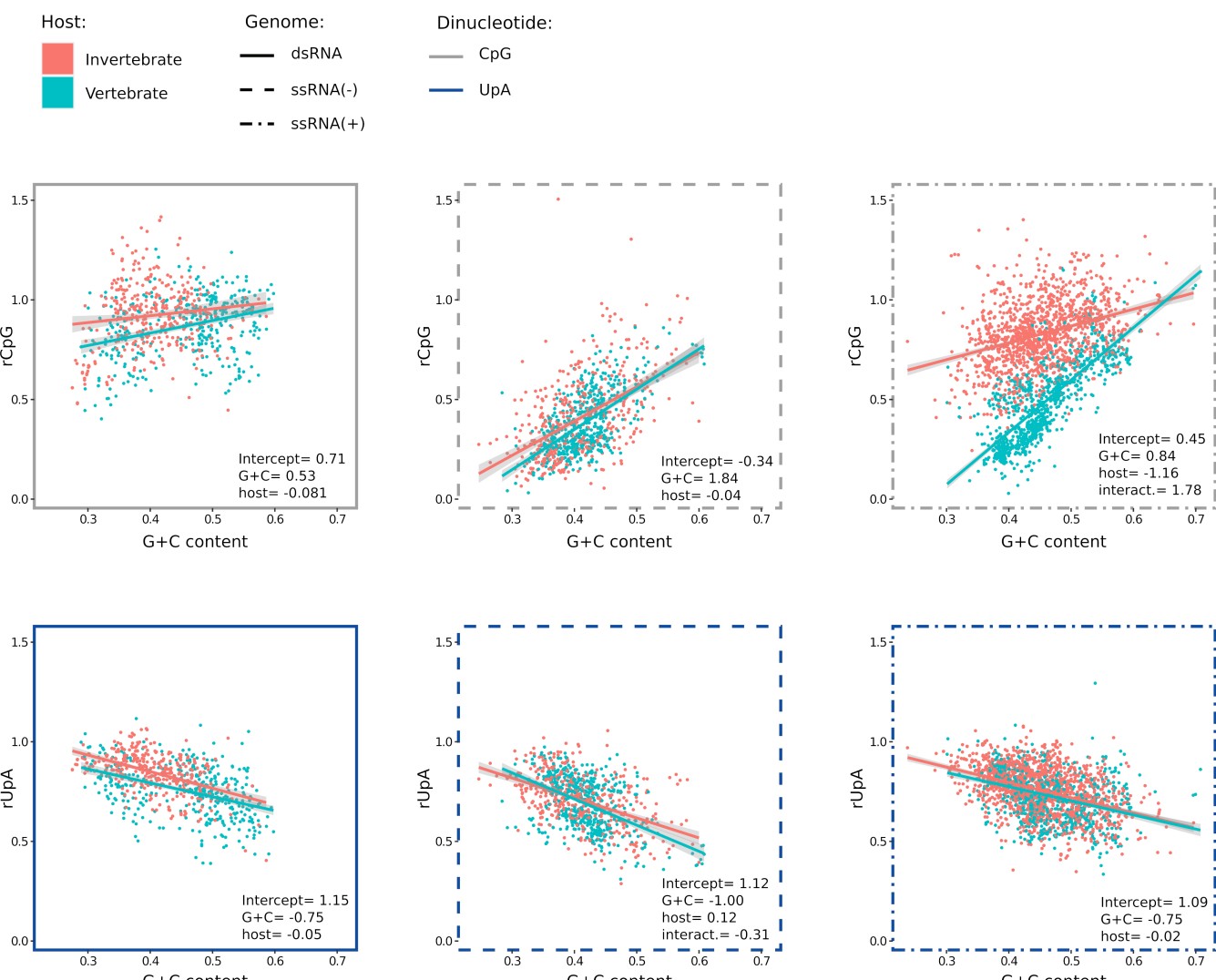

**FIG 2** Ratios of CpG and UpA dinucleotides in viruses with different genome compositions. Ratios are compared between viruses that infect vertebrates or invertebrates. As above, regression lines of rCpG or rUpA on G + C content are shown with confidence intervals. Significant linear model coefficients are reported (full model results and likelihood ratio tests are available in Table S2). The genome composition is denoted by the style of the frames.

contributed to these trends. Following the classification of invertebrate viruses proposed by Shi and co-workers (33), as well as the classification ratified by the ICTV, we restricted the data set to 10 orders that include both vertebrate and invertebrate viruses. For dsRNA viruses we retained two orders (*Durnavirales* and *Reovirales*), three for ssRNA(−) viruses (*Articulavirales*, *Bunyavirales*, and *Jingchuvirales/Mononegavirales*), and five for ssRNA(+) viruses (*Amarillovirales*, *Hepelivirales*, *Nidovirales*, *Picornavirales*, and *Stellavirales*).

For dsRNA viruses, a proper comparison could only be performed for the order *Reovirales*, as only six durnaviruses were included in the vertebrate data set. For invertebrate viruses, no dependency of rCpG on G + C content was evident for durnaviruses, whereas reoviruses showed a clearly positive association (Fig. 3). In the order *Reovirales*, different trends were also observed for vertebrate and invertebrate viruses (Fig. 3).

In the case of ssRNA(−) viruses, limited differences due to host group were observed. Among the three orders, viruses belonging to the *Bunyavirales*, either infecting vertebrates or invertebrates, had the strongest bias against CpG dinucleotides (Fig. 3).

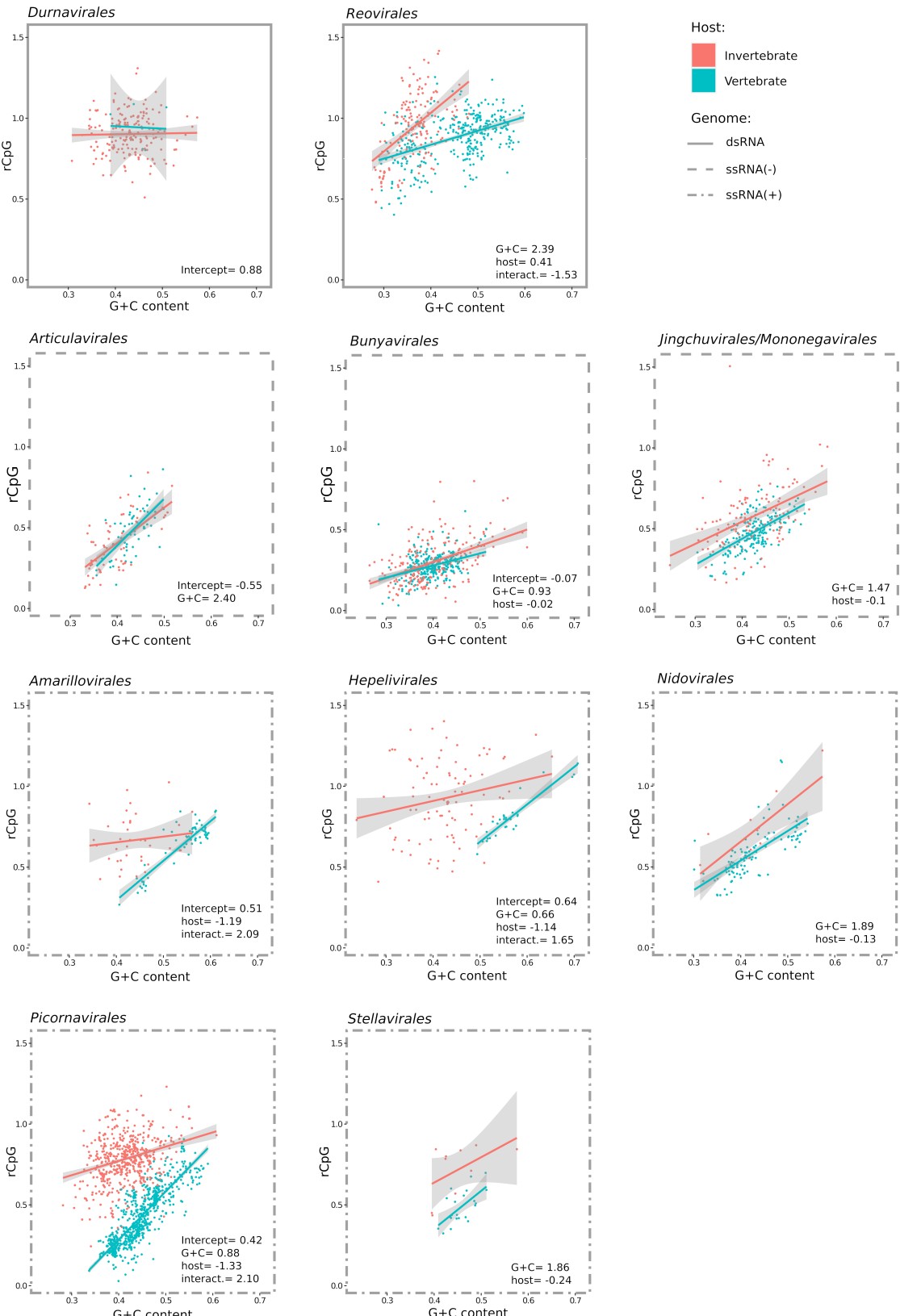

**FIG 3** CpG ratio and G + C content in viruses infecting vertebrates or invertebrates and belonging to different orders. Only viral orders including both viruses that infect vertebrates and invertebrates were analyzed. Regression lines of rCpG on G + C content are shown with confidence intervals. Significant linear model coefficients are reported (full model results and likelihood ratio tests are available in Table S2). The genome composition is denoted by the style of the frames.

The situation was more variegated for ssRNA(+) viruses. The genomes of viruses in the orders *Amarillovirales*, *Hepelivirales*, and *Picornavirales* showed remarkably different patterns depending on the host group. Invertebrate viruses had generally higher rCpG compared to viruses infecting vertebrates and the slopes of the regression lines also differed (Fig. 3). For viruses in these orders, a higher proportion of rCpG variation was explained by G + C content for vertebrate viruses than for invertebrate viruses. The situation was somehow blurred for viruses in the orders *Nidovirales* and *Stellavirales*: invertebrate viruses had, on average, higher rCpG, but the slopes were similar (Fig. 3). It should, however, be noticed that these regressions were performed over few data for invertebrate viruses (12 for *Nidovirales* and 10 for *Stellavirales*).

Overall, these data indicate that, irrespective of the virus order, the host group has a very limited effect on the CpG content of ssRNA(−) viruses. Conversely, the effect is very strong for ssRNA(+) viruses, especially those in the orders *Amarillovirales*, *Hepelivirales*, and *Picornavirales*. In the case of dsRNA viruses, it is more difficult to draw conclusions because the two orders display different patterns, and few vertebrate viruses are available for *Durnavirales*.

## Vertebrate host class influences the CpG content of picornaviruses

The results shown above clearly imply that to further address the role of host classes or phyla in RNA virus dinucleotide composition, virus genome composition and order must be taken into account. We thus decided to focus on picornaviruses because they were abundant in both vertebrate and invertebrate virus data sets. Also, these viruses infect several different hosts. Analyses were performed separately for vertebrate and invertebrate viruses. For the latter, hosts infected by fewer than 10 viruses were removed, leaving four phyla (*Anellida, Arthropoda, Cnidaria,* and *Mollusca*). The regression model showed a clear relationship with G + C content, but no difference among phyla (Fig. 4). In the case of vertebrates, picornaviruses infecting mammals, birds, reptiles, and fish were analyzed. Fish are particularly interesting in this respect because they do not have a ZAP ortholog (9). Besides the clear dependency on G + C content, the regression indicated

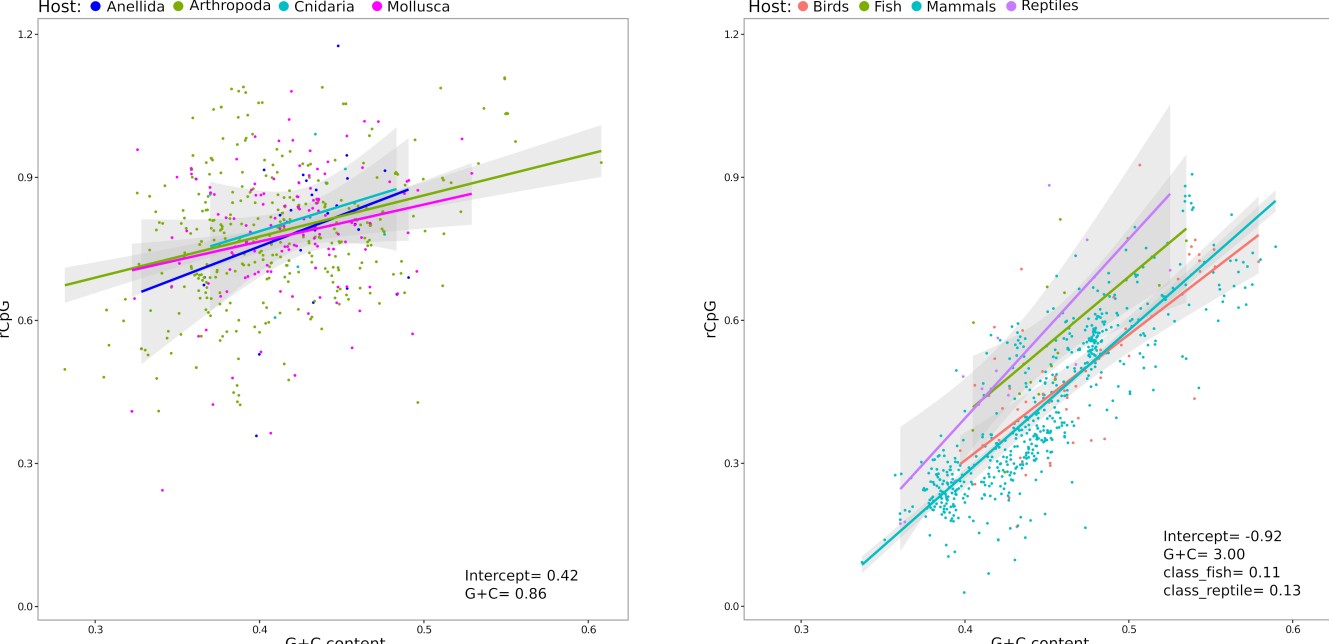

**FIG 4** CpG ratios in picornaviruses infecting different invertebrate and vertebrate hosts. Regression lines of rCpG on G + C content with confidence intervals are shown for picornaviruses that infect invertebrates (left) and vertebrates (right). Significant linear model coefficients are reported (full model results and likelihood ratio tests are available in Table S2). In the case of vertebrate-infecting viruses, comparisons are performed against bird-infecting viruses. Thus, the results of the model indicated that the regression of rCpG on G + C content is not significantly different for mammalian viruses, but it is for fish- and reptile-infecting viruses.

that the rCpG of fish- and reptile-infecting viruses is higher than that of viruses found in mammals and birds. However, the rCpG of fish and reptile viruses is definitely lower than the ones of viruses infecting invertebrates (Fig. 4). We should add that caution is necessary for this analysis, as few fish- and reptile-infecting viruses were available (16 and 14 viruses, respectively).

## CpG dinucleotides are preferentially depleted in A/U-rich contexts in ssRNA(+) and ssRNA(−) viruses

Early observations of mammalian genomes indicated that CpG dinucleotides are particularly depleted in A/T-rich regions (37–39). Moreover, a recent attenuation experiment of HIV-1 and enterovirus A71 (EV-A71, a picornavirus) showed that increasing the number of CpG dinucleotides confers ZAP sensitivity if these dinucleotides are optimally spaced (between 14 and 32 nucleotides) and embedded in an A/U-rich context (20). We thus tested whether the local sequence context of CpG dinucleotides differs in terms of G + C content for RNA viruses. For viruses belonging to different orders, we calculated the average G + C content of 62-nt windows centered on each CpG (i.e., 30 nucleotides upstream and downstream of each CpG, excluding the CpG itself) and compared it to the genome average G + C content.

Results indicated that for most ssRNA(+) and ssRNA(−) viruses, the local C + G content of CpG dinucleotides is significantly higher than the genome average (Fig. 5). Differences were either more evident for vertebrate viruses (*Amarillovirales*, *Hepelivirales*, *Picornavirales*, *Stellavirales*, and *Bunyavirales*) or only significant for these same viruses

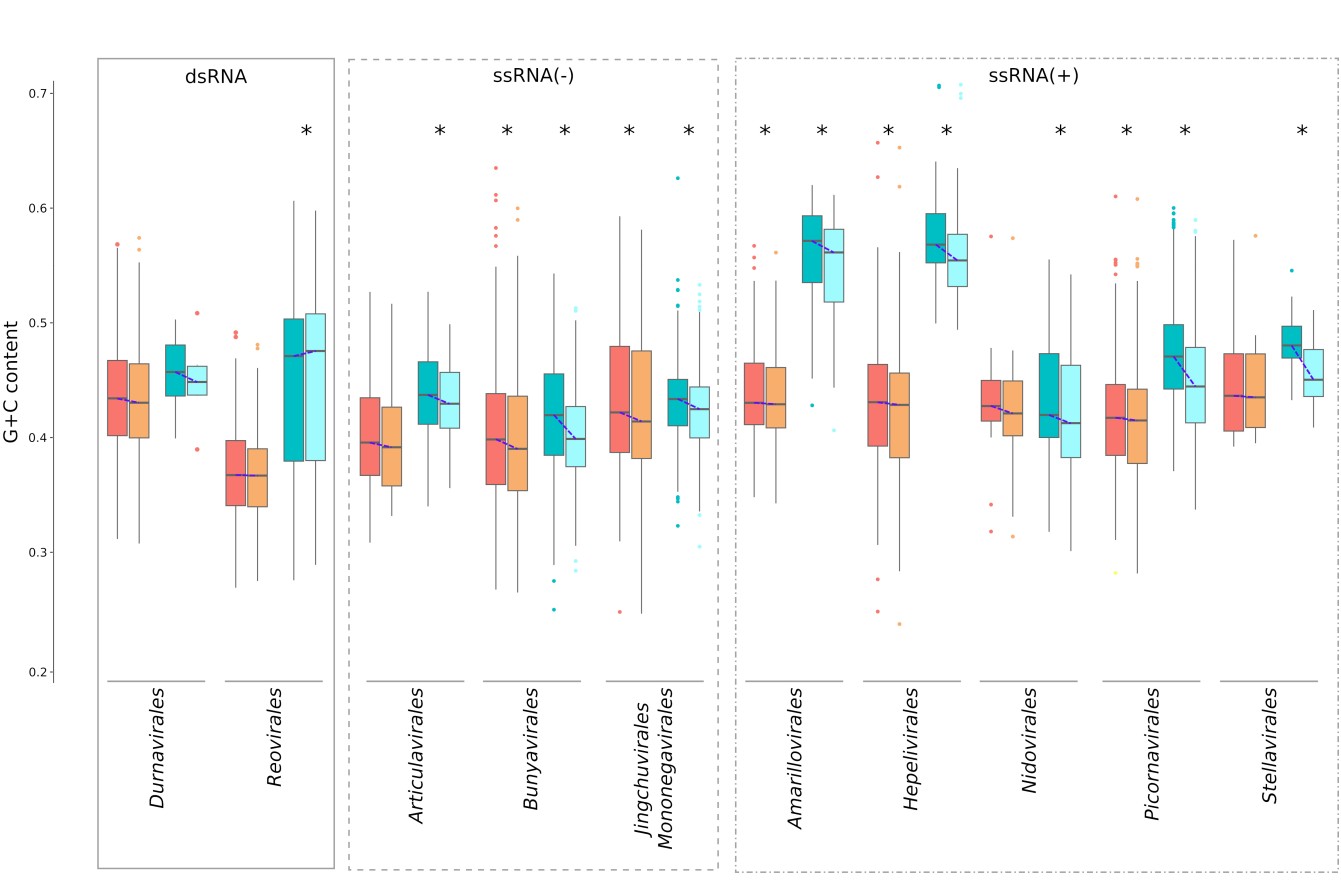

**FIG 5** Analysis of CpG sequence context. The mean G + C content of 62-nt windows centered on each CpG is compared to the genome/genome segment averages. Asterisks denote significant comparisons (*P* < 0.01) after multiple test corrections. Statistical analyses were performed using Wilcoxon rank sum tests for paired samples. Multiple testing was accounted for by Bonferroni correction.

(*Articulavirales* and *Nidovirales*). The most marked differences occurred for vertebrate viruses in the orders *Picornavirales* and *Stellavirales*. Concerning dsRNA viruses, very minor or non-significant differences were observed (Fig. 5). Overall, these results suggest that, in ssRNA viruses, CpG dinucleotides are better tolerated when embedded in G/C-rich contexts.

## CpG depletion varies along viral genomes

We next investigated whether the bias against CpG dinucleotides (as well as its relation to G + C content) differs among viral genomic regions. To this aim, we focused on viruses that infect vertebrates and we selected representative virus families or genera based on their sample size. Thus, we analyzed viruses in the families *Arenaviridae* [ssRNA(−)] and *Picornaviridae* [ssRNA(+)], as well as in the *Orthoreovirus* genus (dsRNA).

The arenavirus genome consists of two or three segments and encodes three or four proteins: nucleoprotein (NP), glycoprotein, RNA-directed RNA polymerase (L), and zinc-binding matrix (Z) protein (Fig. 6A). The latter is very short and not encoded by some arenaviruses that infect reptiles and fish (40). For these reasons, we did not analyze the Z ORF. For the other three ORFs, the linear model underscored significant differences, with NP and L being the most and least CpG-depleted ORFs (Fig. 6A). Notably, in infected cells and mature virions, NP is highly abundant, whereas the least abundant protein is L (41) (https://ictv.global/report/chapter/arenaviridae/arenaviridae).

Picornaviruses have a non-segmented genome with a single ORF that is translated into a polyprotein. Conventionally, the ORF can be divided into three regions: P1 (encoding capsid proteins), P2, and P3 (both encoding non-structural proteins) (Fig. 6B) (https://ictv.global/report/chapter/picornaviridae/picornaviridae). A linear regression model to compare the three regions revealed significant but minor differences among the three regions (Fig. 6B).

Finally, viruses in the *Orthoreovirus* genus have genomes composed of 9–12 segments of linear dsRNA. Most encoded proteins are structural and can be divided based on their localization in the outer or inner capsids (https://ictv.global/report/chapter/spinareoviridae/spinareoviridae/orthoreovirus). The number of non-structural proteins differs in different species (Fig. 6C). We analyzed ORFs encoding proteins of the outer and inner capsids, as well as non-structural proteins. The linear model indicated that ORFs for non-structural proteins are significantly more CpG deleted than ORFs for capsid proteins (Fig. 6C). Limited data on reoviruses indicated that all proteins are expressed at similar levels (42).

## CpG content of coronavirus genomes

Coronaviruses (family *Coronaviridae*, subfamily *Orthocoronavirinae*) belong to the order *Nidovirales*. The emergence of SARS-CoV-2, the causative agent of COVID-19, has spurred efforts to characterize coronavirus genetic diversity, and many viral genomes have been deposited in public repositories. Some of these are still unclassified by the ICTV and were not included in our data set. We thus retrieved additional coronavirus sequences resulting in a final pool of 425 genomes from the four genera (*Alphacoronavirus*, *Betacoronavirus*, *Gammacoronavirus*, and *Deltacoronavirus*) (Table S1). Two-thirds of the coronavirus genome consist of two large ORFs (ORF1a and ORF1b), which encode 16 non-structural proteins. The remaining portion of the genome includes ORFs for the structural proteins, namely spike (S), envelope (E), membrane (M), and nucleoprotein (N). A variable number of accessory proteins are also encoded by coronaviruses (43). Because of their small sizes, accessory ORFs and the E ORF were not analyzed (Fig. 6D). We thus compared the CpG content in ORF1a and ORF1b, S, M, and N using linear models. The model without interaction showed a better fit to the data and indicated that different ORFs have a distinctive bias against CpG, with the N ORF being the most depleted, followed by S, ORF1a/ORF1b, and M (Fig. 6D).

SARS-CoV-2 is a member of the *Sarbecovirus* subgenus (genus *Betacoronavirus*), and closely related viruses were identified in bats and pangolins (44–46). A previous study

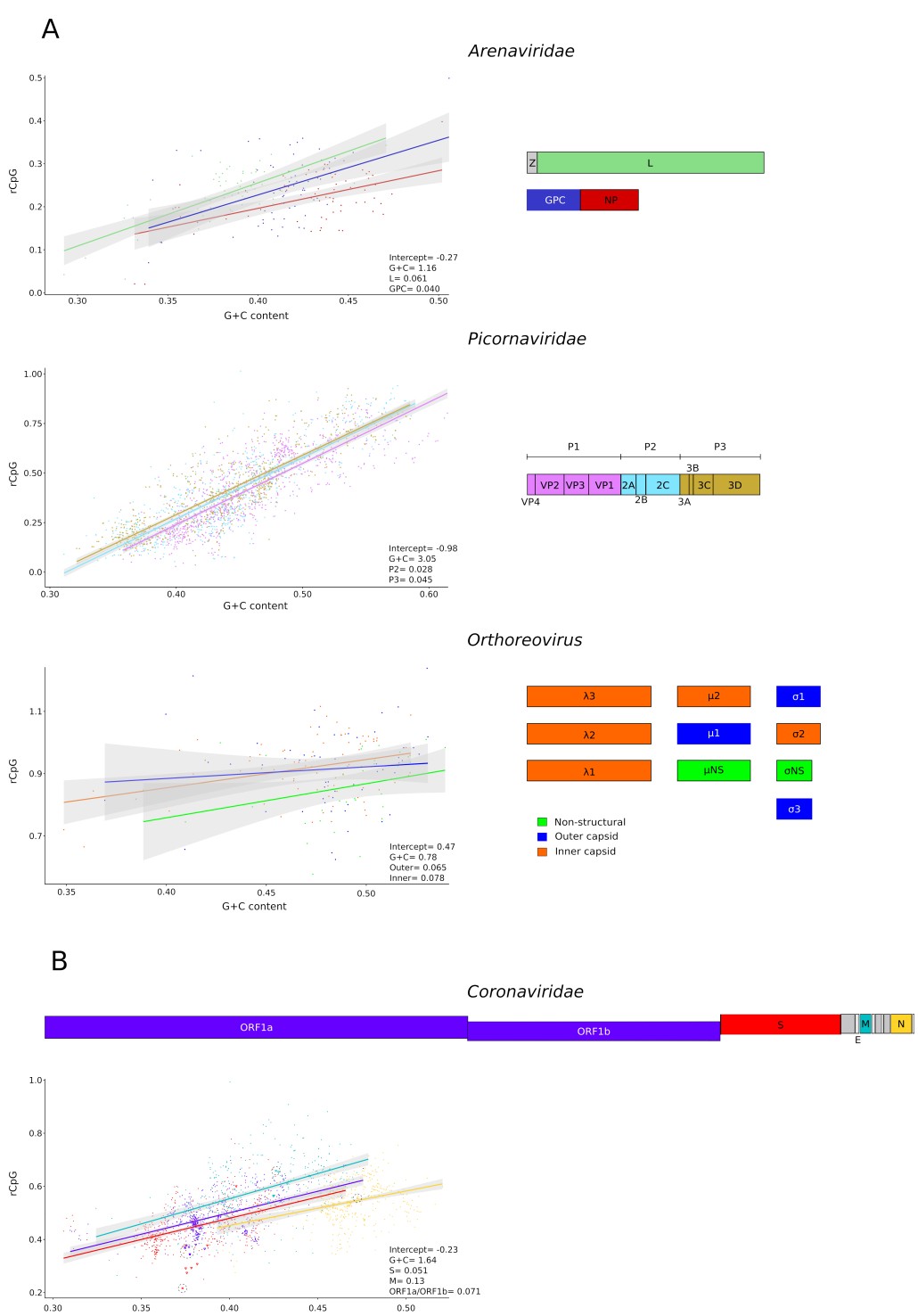

**FIG 6** Comparisons among viral ORFs. (A) Regression of rCpG on G + C content (left) for different ORFs in the genomes of arenaviruses, picornaviruses, and reoviruses. Significant linear model coefficients are reported (full model results and likelihood ratio tests are available in Table S2). On the right, representative genomes are drawn to scale with ORFs colored as in the plots. (B) Comparisons among ORFs from the genomes of 425 coronaviruses. A schematic representation of the SARS-CoV-2 genome is also drawn to scale. Data in the plot are colored as in the scheme. Data points corresponding to viruses closely related to SARS-CoV-2 are shown as triangles, the ones corresponding to SARS-CoV-2 as crosses. The latter are also circled. As in panel (A), significant linear model coefficients are reported (full model results and likelihood ratio tests are available in Table S2).

indicated that an adaptive shift favored a depletion of CpG dinucleotides in the ORF1a/ORF1b region of SARS-CoV-2 and related bat/pangolin viruses (47). By analyzing rCpG as a function of G + C content, we confirm that, in these viruses, ORF1a and ORF1b tend to be particularly CpG-poor. An even stronger depletion is observed for the S ORF, with the rCpG value for SARS-CoV-2 being the lowest among coronaviruses. However, no unusual depletion was observed for the N and M ORFs of SARS-CoV-2 and related sarbecoviruses (Fig. 6D).

## CpG dinucleotides are poorly conserved

Finally, we investigated whether CpG dinucleotides show unusual conservation across phylogenies of orthologous viral genes. To this aim, we focused on the L gene of viruses in the *Mammarenavirus* genus (family *Arenaviridae*) and on the M protein of viruses in the *Betacoronavirus* genus (family *Coronaviridae*). These genera were selected as representative of ssRNA(−) and RNA(+) viruses. The L and M ORFs show the highest CpG content in the respective genomes (Fig. 6A and D). We thus generated nucleotide alignments and we counted the fraction of sequences sharing each CpG dinucleotide. As a comparison, the same procedure was applied to GpC dinucleotides. Results indicated that CpG dinucleotides are significantly less conserved than GpC dinucleotides, both in the mammarenavirus L gene and in the betacoronavirus M gene (Fig. 7). In the L gene, no difference in conservation was observed among regions that encode or do not encode known protein domains (Fig. 7). Overall, these results indicate that CpG dinucleotides are either lost by mutational biases or selected against in these viral genes.

## DISCUSSION

The identification of ZAP as a restrictor of CpG-enriched viruses provided an elegant explanation for the observation that RNA viruses that infect vertebrates are CpG-depleted (12). Whereas the depletion of CpG dinucleotides in vertebrate genomes is considered to derive, at least partially, from methylation, RNA is not a substrate for methyl transferases. Thus, CpG suppression in vertebrate-infecting viruses is generally considered as a mimicry of the dinucleotide composition of the host to avoid detection. In line with this view, the experimental introduction of CpG dinucleotides in both ssRNA(+) and ssRNA(−) viral genomes was shown to restrict replication in a ZAP-dependent manner (15–21, 31, 35). Conversely, most invertebrate genomes display no or low-level cytosine methylation and little bias against CpG dinucleotides (3, 5–9). Also, invertebrates do not express ZAP, which evolved in tetrapods by gene duplication (9). As a consequence, invertebrate-infecting viruses would not be expected to suppress CpG dinucleotides in their genomes. This expectation was previously tested in flaviviruses (order *Amarillovirales*), which include arthropod-borne viruses (which infect both vertebrates and invertebrates, e.g., ZIKV), members with no known vector (NKVF), and insect-specific flaviviruses. The latter was shown to display a definitely higher CpG content than arthropod-borne flaviviruses and NKVF (23, 26). Consistently, increasing the number of CpG dinucleotides in the ZIKV genome curtails viral replication in vertebrates, but not in mosquito cells (26).

Other observations, however, suggest that additional mechanisms might contribute to generate a more complex scenario. First, vertebrate mitochondrial genomes, which are not methylated, are also CpG-depleted (3). Second, an analysis of RNA virus genomes, although unaware of variation in G + C content, indicated that viral taxonomy is an important determinant of CpG bias, whereas host association is less relevant (32). We thus performed a comprehensive analysis of RNA virus genomes to identify the determinants of biases against CpG (and UpA) dinucleotides. Our results indicate that whereas UpA dinucleotides tend to be similarly under-represented in viruses with different genome compositions, which infect vertebrates or invertebrates, important differences are observed for CpG. In general, the tendency for vertebrate-infecting viruses to have stronger CpG bias than invertebrate-infecting viruses is not universal. Rather, it is mainly driven by ssRNA(+) viruses and, to a lesser extent, by dsRNA viruses. Conversely, ssRNA(−)

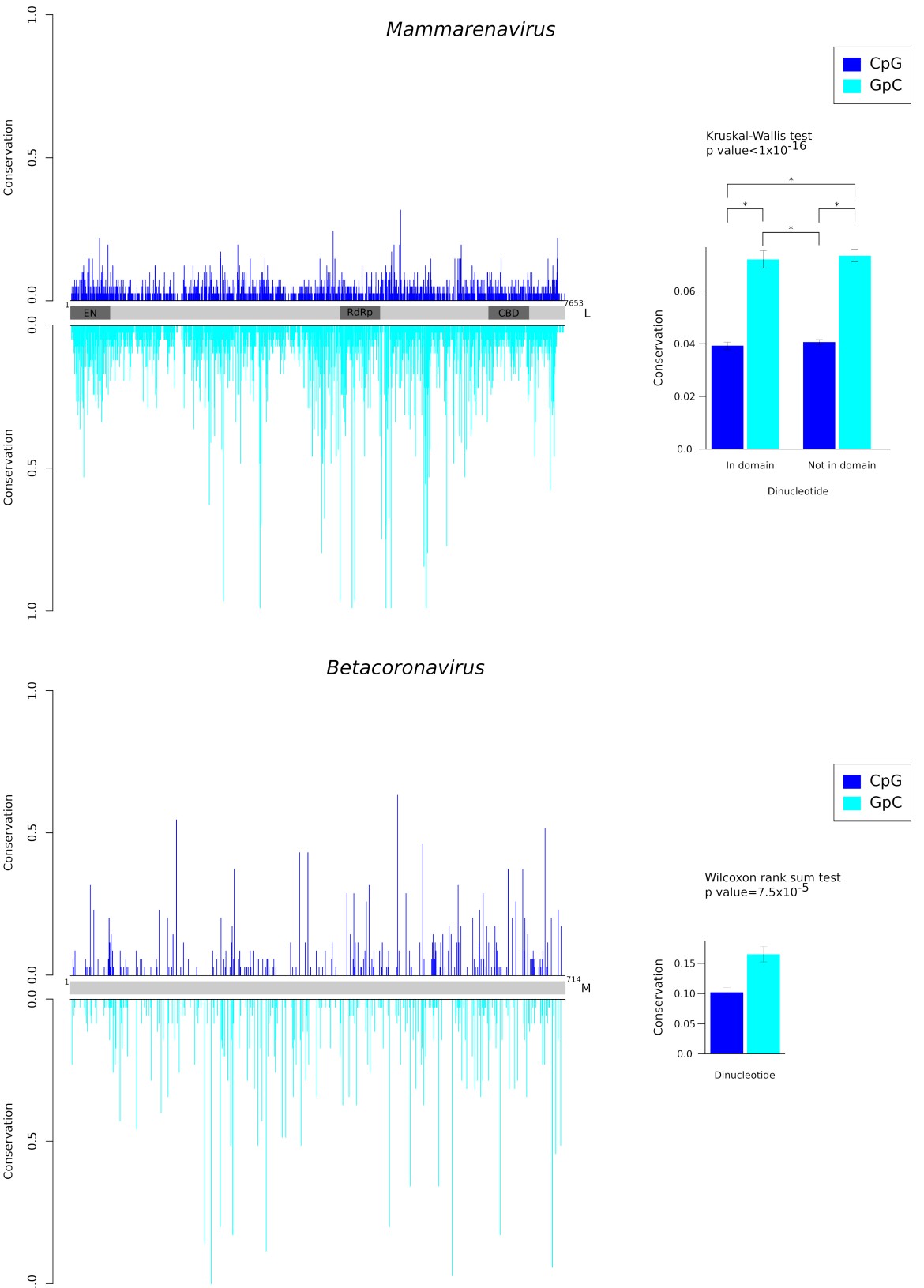

**FIG 7** Conservation of CpG dinucleotides. The conservation (fraction of sequences sharing each dinucleotide) of CpG and GpC dinucleotides is plotted along the sequence alignment of the mammarenavirus L gene (upper panel) and of the betacoronavirus M ORF (lower panel). In the L gene, regions encoding known functional domains are shown (EN, endonuclease, RdRp, RNA-dependent RNA polymerase, and CBD, cap-binding domain). The barplots represent the average conservation with standard errors. Asterisks denote pairwise significant ($P < 0.0001$) comparisons at the Nemenyi *post hoc* test.

viruses have a dinucleotide composition that is largely unrelated to the host. Also, these viruses, and especially those in the order *Bunyavirales*, are extremely CpG-depleted. This observation is quite puzzling, as the bias against CpG dinucleotides in invertebrate ssRNA(−) viruses cannot be driven by the selective pressure imposed by ZAP, nor by any form of mimicry, as most invertebrate genomes are not CpG-depleted (3, 5–9). For this very reason, it is difficult to envisage that, in the cells of invertebrates, some unknown mechanism targets CpG-rich RNA, as this would affect host mRNAs, as well. We thus suggest that CpG depletion in ssRNA(−) viruses is related to some aspects of virus biology (see below).

In the case of ssRNA(+) viruses, differences between invertebrate and vertebrate hosts were observed, although viruses in different orders had distinct trends. Because they were abundant in our data set, we focused on viruses in the order *Picornavirales* to investigate whether host class/phylum influenced CpG content. We were particularly interested in the comparison of viruses that infect fish (that do not express ZAP) with those that infect other vertebrates. Fish picornaviruses did have a weaker bias against CpG than picornaviruses infecting birds or mammals. However, their rCpG was comparable to that of picornaviruses infecting reptiles (which have a functional ZAP ortholog) and, in any case, definitely lower than that of picornaviruses infecting invertebrates. Whereas these data will require additional validation due to the small sample size of fish- and reptile-infecting picornaviruses, they suggest that ZAP may have a role as a driver of CpG content in ssRNA(+) viruses, but additional mechanisms must contribute. A similar conclusion was recently reached by Odon and co-workers, who found considerable variability in ZAP activity among bird species, with no parallel in the diversity of CpG content of host transcriptomes and RNA viral genomes (35).

These observations are not in contrast with experimental evidence that artificially increasing the number of CpG dinucleotides in the genome of EV-A71 promotes viral restriction by ZAP (20, 35). Indeed, the fact that CpG content is determined by multiple factors does not imply that ZAP is ineffective. Also, recent evidence indicated that the sheer number of CpG dinucleotides is not sufficient to promote restriction by ZAP. The spacing and sequence context of CpG dinucleotides play an important role. Specifically, mutant HIV-1 and EV-A71 viruses display the highest sensitivity to ZAP when CpG dinucleotides are inserted in A/U-rich regions (20). This prompted us to analyze the local G + C content of CpG dinucleotides in viral genomes. Our data indicate that, with the exclusion of dsRNA viruses, for viruses in most orders, the CpG contexts are less A/U-rich than the genome average. However, this was observed for both viruses that infect vertebrates and for those that infect invertebrates, although effects were consistently weaker for the latter. Again, these observations point to a role for ZAP, but also to the contribution of other factors.

One interesting possibility is that CpG dinucleotides in A/U-rich context have a general negative effect on RNA stability or metabolic processes (e.g., on the efficiency of transcription/replication or translation). This would explain why some vertebrate interferon genes, which need to be rapidly induced and expressed at high levels during viral infection, are particularly depleted in CpG dinucleotides (35). This would also reconcile the observation of CpG depletion in vertebrate mitochondrial genomes, but not the fact that invertebrate transcriptomes have normal representation of CpG dinucleotides.

Following these lines of thought, we tested whether, within the same viral genomes, individual viral ORFs display different CpG content. In particular, we hypothesized that ORFs that encode abundant proteins are particularly CpG-depleted. To test whether this is the case, we analyzed viral families or genera with distinct genome composition and organization. Overall, we obtained mixed evidence. Consistent with a possible bias proportional to protein abundance, we observed greater depletion in the NP ORF of arenaviruses and lesser in the L ORF (41). Likewise, the N protein of coronaviruses is the most depleted in CpG dinucleotides as well as the most abundant during infection. However, M is expressed at higher levels than S and ORF1a/ORF1b but shows a

weaker bias against CpG (48, 49). Finally, the little differences among ORFs observed in picornaviruses might be expected on the basis that the single ORF is translated as a polyprotein and subsequently cleaved, whereas for reoviruses limited experimental evidence suggests that all proteins are expressed at similar levels (42). Thus, it is difficult to draw definite conclusions about why individual ORFs differ in CpG content. Despite such differences, analysis of two viral genes, the L gene of mammarenaviruses and the M ORF of betacoronaviruses, indicated that, in both cases, CpG dinucleotides are significantly less conserved than GpC dinucleotides. In the L gene, analysis of regions that encode known functional domains revealed no difference in conservation compared to regions outside of such domains. Thus, CpG conservation (or lack thereof) does not seem to be related to functional constraints. Rather, these results suggest that depletion of CpG dinucleotides results from either mutational biases or from a selective pressure against their presence in viral genes, irrespective of their location. Additional analyses of mutation spectra and selective patterns will be necessary to disentangle these possibilities.

As a final note, we wish to add that, as mentioned above, a previous study suggested that adaptive depletion of CpG dinucleotides occurred in the lineage of SARS-CoV-2 and related viruses (47). The authors specifically analyzed ORF1ab and, whereas they did not factor G + C content in, they used a phylogenetic comparative method to infer on which branch(s) of the phylogeny the adaptive shift occurred. Herein, we confirm that, among coronaviruses, SARS-CoV-2 and related sarbecoviruses have a remarkable depletion of CpG dinucleotides in ORF1ab. On one hand, an even stronger depletion is observed for the S ORF, for which SARS-CoV-2 displayed the lowest value. On the other hand, the rCpG of SARS-CoV-2 N and M ORFs is not particularly low. Because coronaviruses recombine rampantly, the different trends of these ORFs might derive from their evolutionary history and acquisition from different parental genomes. Whatever the underlying reason for the observed heterogeneity in CpG content, we caution against the interpretation that an adaptive shift for CpG depletion in SARS-CoV-2 and related sarbecoviruses occurred as a strategy to evade innate immunity effectors. If this were the case, a more generalized depletion would be expected. Also, the results we present here do not strongly support the role of the host innate immune system (i.e., ZAP) as a major driver of CpG content (47).

In summary, both for SARS-CoV-2 and, more generally, for RNA viruses that infect vertebrates and invertebrates, a number of questions remain unanswered and the ultimate mechanisms underlying CpG depletion are obscure. Our data underscore important differences among viruses with different genome compositions and belonging to different orders. Thus, additional analyses are warranted, not only to gain a better understanding of viral evolution but also to evaluate the portability of approaches based on the modulation of CpG content as a strategy for vaccine development.

## MATERIALS AND METHODS

### Virus data set

A list of exemplar viruses was retrieved from the ICTV Virus Metadata Resource (https://ictv.global/vmr). This list includes information regarding virus name, host source, isolate designation, genome composition, and the GenBank accession number for the genomic sequence. We only retained RNA viruses that infect vertebrates, with the exclusion of reverse-transcribing viruses. All entries were manually inspected to further purge viruses that infect vertebrates but are transmitted by arthropods or other invertebrates. The final list included 1,836 genomes or genome segments. In addition, we retrieved a list of viruses infecting invertebrate hosts from a previous study (33). We created two datasets with information regarding viral taxonomy classification, genome composition, host, genome/genomic segment sequence, and C + G content.

Finally, genomes/genome segments with a sequence length shorter than 500 nucleotides were excluded. Overall, the data set included 4,144 sequences (Table S1).

Representative coronavirus species were retrieved from a previous study (50) and integrated with the exemplar species already present in the ICTV VMR data set. Briefly, viruses showing less than 99% sequence identity were retained (Table S1). Non-structural and structural ORFs longer than 500 nucleotides were retained (e.g., the envelope protein was excluded from the analysis).

## Dinucleotide observed/expected ratio

To investigate dinucleotide biases, we calculated the observed/expected ratio for CpG, GpC, ApU, and UpA dinucleotides. Specifically, the frequency of each dinucleotide in each genome/genomic segment (i.e., the observed frequency) was divided by the product of the frequencies of each contributing nucleotide (i.e., the expected frequency). Thus, for CpG, we calculated the number of CpG along the genome divided by the number of all possible dinucleotides; this frequency was then divided by the product of C and G frequencies. All analyses were performed in the R environment.

## C + G context of CpG dinucleotides

We retrieved the genomic position of all CpG dinucleotides in each viral species and calculated the G + C content in a window of 62 nucleotides centered on the CpG. Specifically, we selected 30 nucleotides upstream and downstream of each CpG (excluding the dinucleotide itself) and calculated the G + C content. This window span was chosen because it is the optimal distance between two CpG dinucleotides for ZAP sensitivity (20). We then compared this value with the average G + C content calculated for the whole genome/genomic sequence of that specific viral species.

Statistical significance was calculated by using Wilcoxon Rank Sum tests for paired samples. Bonferroni correction was applied to account for multiple testing. All analyses were performed in the R environment.

## Conservation of CpG dinucleotides

To quantify the conservation of CpG dinucleotides, we analyzed the betacoronavirus M gene and the mammarenavirus L gene. Using the R package ape, for each gene, we selected sequences with an overall nucleotide identity lower than 90%. We obtained a set of 35 betacoronavirus and 41 mammarenavirus sequences. These were aligned using the Revtrans utility (51) with MAFFT as an aligner (52). We then counted the number of CpG dinucleotides along the alignment and how many sequences carried each dinucleotide. As a comparison, the same procedure was applied to GpC dinucleotides. Statistical significance was evaluated by Wilcoxon rank sum test or Kruskal-Wallis test. Pairwise comparisons were performed with the Nemenyi *post hoc* test using the PMCMRplus R package.

## Statistical analysis

We linearly modeled the relationship between CpG (UpA) ratios and G + C content including a term corresponding to a grouping variable such as host group, host phylum, host class, or viral ORF. We fitted each model both by including and not including the interaction between the C + G and the grouping variable. We used the likelihood ratio goodness-of-fit test to compare the two fittings: the one including C + G and grouping variable interaction was selected with a 0.05 significance level. Model parameters were considered significant if their Wald test *P* value was less than 0.05.

### ACKNOWLEDGMENTS

This work was supported by the Italian Ministry of Health ("Ricerca Corrente 2023" to M.S.).

## AUTHOR AFFILIATIONS

[1]Bioinformatics Lab, Scientific Institute IRCCS E. MEDEA, Bosisio Parini, Italy
[2]Department of Physiopathology and Transplantation, University of Milan, Milan, Italy
[3]Don C. Gnocchi Foundation ONLUS, IRCCS, Milan, Italy

## AUTHOR ORCIDs

Diego Forni http://orcid.org/0000-0001-9291-5352
Uberto Pozzoli http://orcid.org/0000-0003-0670-7106
Rachele Cagliani http://orcid.org/0000-0003-2670-3532
Manuela Sironi http://orcid.org/0000-0002-2267-5266

## FUNDING

| Funder | Grant(s) | Author(s) |
|--------|----------|-----------|
| Ministero della Salute (Italy Ministry of Health) | Ricerca Corrente 2023 | Manuela Sironi |

## AUTHOR CONTRIBUTIONS

Diego Forni, Conceptualization, Data curation, Formal analysis, Investigation, Methodology, Writing – original draft | Uberto Pozzoli, Conceptualization, Data curation, Formal analysis, Investigation, Methodology | Rachele Cagliani, Data curation, Formal analysis, Investigation | Mario Clerici, Data curation, Investigation, Writing – original draft | Manuela Sironi, Conceptualization, Data curation, Formal analysis, Funding acquisition, Investigation, Methodology, Supervision, Writing – original draft, Writing – review and editing

## ADDITIONAL FILES

The following material is available online.

### Supplemental Material

**Supporting Table S1 (Spectrum02529-23-s0001.xls).** List of viral genomes/genome segments used in this study.
**Supporting Table S2 (Spectrum02529-23-s0002.xls).** Results of the likelihood ratio tests (LRTs) and of linear models.

### Open Peer Review

**PEER REVIEW HISTORY (review-history.pdf).** An accounting of the reviewer comments and feedback.

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
