## [Reviewer comments · Microbiology Spectrum]

Microbiology Spectrum

Dinucleotide biases in RNA viruses that infect vertebrates or invertebrates

Diego Forni, Uberto Pozzoli, Rachele Cagliani, Mario Clerici, and Manuela Sironi

Corresponding Author(s): Manuela Sironi, Scientific Institute IRCCS E. Medea

Review Timeline:

Submission Date:	June 16, 2023
Editorial Decision:	July 23, 2023
Revision Received:	July 27, 2023
Accepted:	August 12, 2023

Editor: Takamasa Ueno

Reviewer(s): The reviewers have opted to remain anonymous.

Transaction Report:

DOI: <https://doi.org/10.1128/spectrum.02529-23>

July 23, 2023

Dr. Manuela Sironi
Scientific Institute IRCCS E. Medea
Bioinformatics
Via Don L. Monza 20
Bosisio Parini
Italy

Re: Spectrum02529-23 (**Dinucleotide biases in RNA viruses that infect vertebrates or invertebrates**)

Dear Dr. Manuela Sironi:

Link Not Available

Sincerely,

Takamasa Ueno

Journals Department
Reviewer comments:

Reviewer #1 (Comments for the Author):

The analysis could be completed with more phylogenetic analyses

Reviewer #2 (Comments for the Author):

Forni and colleagues comprehensively analyzed RNA viruses that infect vertebrates and/or invertebrates to determine biases for CpG and UpA dinucleotide contexts. They showed vertebrate-infecting viruses, particularly single-stranded, plus-stranded RNA viruses, have relatively stronger CpG bias. Interestingly, the CpG ratio is variable in each virus as shown in Fig. 3. It is also

interesting that SARS-CoV-2 has different GC contents among S, N and M genes. However, the authors should address the following concerns to fortify the manuscript.

Major concerns;

1. Line 33, an immune evasion strategy. This is for an evasion strategy against ZAP. However, ZAP is an effector of the "innate" immune response against virus infection. The same thing is found lines 363 and 365. These (and more if there are) are required to be revised.

2. Figs. 2 and 3. You should mention dsRNA, ssRNA (-) and ssRNA (+) like Fig. 5. It is hard to understand which virus the authors show.

Minor concerns;

Line 124, "that". Is this "than"?

Staff Comments:

Preparing Revision Guidelines

Please return the manuscript within 60 days; if you cannot complete the modification within this time period, please contact me. If you do not wish to modify the manuscript and prefer to submit it to another journal, please notify me of your decision immediately so that the manuscript may be formally withdrawn from consideration by Microbiology Spectrum.

Dinucleotides biases in RNA viruses that infect vertebrates or invertebrates.

In this article Forni et al analyse the CpG and UpA dinucleotide usage in various RNA viruses and conclude that ZAP can not be the sole agent responsible for the suppression of CpG.

This article follows upon the 2013 article by Simmonds where it was clearly showed that invertebrates do not suppress CpG and therefore viruses that infect them also tend to have a higher CpG content. I think it would have been nice to start from there and mention this article in more details, because the proposed publication by Forni et al. follows directly upon it.

I like that it goes in details for each viral family but where I think the authors could elaborate further is in the analyses of the CpG rich regions and include an analysis of the conservation of this sequence in the viral families. Certainly their conclusion suggests this and a deeper analysis of these sequences: are they conserved domains of important proteins? Or perhaps the CpG here contributes to overall structure, would make for a stronger argument. To me without an analysis of this kind the article concludes little more than what was already established by Simmonds in 2013.

About ZAP, at the moment we do not know what exactly triggers ZAP, there are abundant information to show CpG does but it is not the only motif, and if we look at transcriptome of even mammalian cells, it is clear that some transcripts are CpG rich yet they do not seem to induce ZAP or maybe they do and we do not have evidence for this yet. So it is difficult to ascertain anything regarding binding to ZAP because so much is missing. Which is why if the authors want to claim that CpG patterns in viruses are not entirely linked to ZAP activity they must strengthen their argument in favour of phylogeny with more analyses.

Positive: I think the article is clearly written, the analyses are correct.

Minor correction: increase the font of the Y axis so that we can see straight away if we look at CpG or UpA or find another way to add this in the legend so that it is more evident.

Forni and colleagues comprehensively analyzed RNA viruses that infect vertebrates and/or invertebrates to determine biases for CpG and UpA dinucleotide contexts. They showed vertebrate-infecting viruses, particularly single-stranded, plus-stranded RNA viruses, have relatively stronger CpG bias. Interestingly, the CpG ratio is variable in each virus as shown in Fig. 3. It is also interesting that SARS-CoV-2 has different GC contents among S, N and M genes. However, the authors should address the following concerns to fortify the manuscript.

Major concerns;

1. Line 33, an immune evasion strategy. This is for an evasion strategy against ZAP. However, ZAP is an effector of the “innate” immune response against virus infection. The same thing is found lines 363 and 365. These (and more if there are) are required to be revised.
2. Figs. 2 and 3. You should mention dsRNA, ssRNA (-) and ssRNA (+) like Fig. 5. It is hard to understand which virus the authors show.

Minor concerns;

Line 124, “that”. Is this “than”?

Reviewer #1

In this article Forni et al analyse the CpG and UpA dinucleotide usage in various RNA viruses and conclude that ZAP can not be the sole agent responsible for the suppression of CpG.

This article follows upon the 2013 article by Simmonds where it was clearly showed that invertebrates do not suppress CpG and therefore viruses that infect them also tend to have a higher CpG content. I think it would have been nice to start from there and mention this article in more details, because the proposed publication by Forni et al. follows directly upon it.

>>> RE: We are grateful to the Reviewer for their comments on our manuscript and for thoughtful suggestion. We agree that our manuscript builds on data presented by Simmonds and co-workers, although it reaches different conclusions. As suggested, we have now mentioned Simmonds' work in more detail, as follows:

“For instance, Simmonds and co-workers analyzed the representation of CpG dinucleotides in the genomes of RNA and small DNA viruses that infect mammals and insects (which do not possess ZAP) (7). They found no CpG depletion among insect viruses. Conversely, mammalian RNA viruses with single stranded genomes and reverse transcribing viruses, but not dsRNA viruses, showed CpG suppression. Specifically, CpG depletion in these viruses was related to the G+C composition of their genomes. The authors thus concluded that mammal-infecting RNA viruses that expose their genetic material to the cytoplasm are subject to selection against CpG”.

I like that it goes in details for each viral family but where I think the authors could elaborate further is in the analyses of the CpG rich regions and include an analysis of the conservation of this sequence in the viral families. Certainly their conclusion suggests this and a deeper analysis of these sequences: are they conserved domains of important proteins? Or perhaps the CpG here contributes to overall structure, would make for a stronger argument. To me without an analysis of this kind the article concludes little more than what was already established by Simmonds in 2013.

>>> RE: Thank you for raising this interesting point. We agree that an analysis of CpG conservation across viral gene phylogenies is a good strategy to obtain further insight. We thus selected two viral genera (*Mammarenavirus*, ssRNA(-) and *Betacoronavirus*, ssRNA(+)) and the two viral genes showing the highest CpG content in the respective genomes. We generated nucleotide alignments and we counted the fraction of sequences sharing each CpG dinucleotide. As a comparison, the same procedure was applied to GpC dinucleotides. Results indicated that CpG dinucleotides are significantly less conserved than GpC dinucleotides both in the mammarenavirus L gene and in the betacoronavirus M gene. In the L gene, we checked for differences among regions that encode or do not encode known protein domains. Overall, we conclude that CpG dinucleotides are either lost by mutation biases or selected against in these viral genes, irrespective of their location. The results, discussion and methods were updated to include these data, which are summarized in Figure 7.

About ZAP, at the moment we do not know what exactly triggers ZAP, there are abundant information to show CpG does but it is not the only motif, and if we look at transcriptome of even mammalian cells, it is clear that some transcripts are CpG rich yet they do not seem to induce ZAP or maybe they do and we do not have evidence for this yet. So it is difficult to ascertain anything regarding binding to ZAP because so much is missing. Which is why if the authors want to claim that CpG patterns in viruses are not entirely linked to ZAP activity they must strengthen their argument in favour of phylogeny with more analyses.

>>> RE: Thank you so much for this comment. We fully agree that we still miss many details about the function of ZAP and the mechanisms underlying its binding and induction. Nonetheless, we consider that, whatever its binding specificity, restriction by ZAP cannot explain CpG depletion in organisms (invertebrates and fish) that possess no ZAP ortholog. For instance, the binding specificity of ZAP

cannot explain why bunyaviruses that infect vertebrates and invertebrates are similarly CpG depleted and why picornaviruses infecting fish and reptiles have similar CpG representation. This said, we have now included additional analyses of CpG (and GpC) conservation across the phylogenies of mammarenaviruses and betacoronaviruses.

Positive: I think the article is clearly written, the analyses are correct.

>>> RE: Thank you so much for your appreciation of our work

Minor correction: increase the font of the Y axis so that we can see straight away if we look at CpG or UpA or find another way to add this in the legend so that it is more evident.

>>> RE: We have increased the Y axis font. In figures 1 and 2 we have denoted CpG plots with a black frame and UpA plots with a blue frame.

Reviewer #2 (Comments for the Author):

Forni and colleagues comprehensively analyzed RNA viruses that infect vertebrates and/or invertebrates to determine biases for CpG and UpA dinucleotide contexts. They showed vertebrate-infecting viruses, particularly single-stranded, plus-stranded RNA viruses, have relatively stronger CpG bias. Interestingly, the CpG ratio is variable in each virus as shown in Fig. 3. It is also interesting that SARS-CoV-2 has different GC contents among S, N and M genes. However, the authors should address the following concerns to fortify the manuscript.

Major concerns;

1. Line 33, an immune evasion strategy. This is for an evasion strategy against ZAP. However, ZAP is an effector of the "innate" immune response against virus infection. The same thing is found lines 363 and 365. These (and more if there are) are required to be revised.

>>> RE: We are grateful to the Reviewer for their comments on our manuscript and for thoughtful suggestion. We apologize for the imprecise wording. We have now modified the sentences to make it clear that we are referring to innate immune responses.

2. Figs. 2 and 3. You should mention dsRNA, ssRNA (-) and ssRNA (+) like Fig. 5. It is hard to understand which virus the authors show.

>>> RE: Thank you for this observation. In figures 2 and 3, genome composition is coded by the style of the frame, as per legend.

Minor concerns;

Line 124, "that". Is this "than"?

>>> RE: The error was corrected. Thank you.

August 12, 2023

Dr. Manuela Sironi
Scientific Institute IRCCS E. Medea
Bioinformatics
Via Don L. Monza 20
Bosisio Parini
Italy

Re: Spectrum02529-23R1 (**Dinucleotide biases in RNA viruses that infect vertebrates or invertebrates**)

Dear Dr. Manuela Sironi:

Your manuscript has been accepted, and I am forwarding it to the ASM Journals Department for publication. You will be notified when your proofs are ready to be viewed.

Sincerely,

Takamasa Ueno
Editor, Microbiology Spectrum
